# Involvement of *N*-glycan in Multiple Receptor Tyrosine Kinases Targeted by Ling-Zhi-8 for Suppressing HCC413 Tumor Progression

**DOI:** 10.3390/cancers11010009

**Published:** 2018-12-21

**Authors:** Ren-In You, Wen-Sheng Wu, Chuan-Chu Cheng, Jia-Ru Wu, Siou-Mei Pan, Chi-Wen Chen, Chi-Tan Hu

**Affiliations:** 1Department of Laboratory Medicine and Biotechnology, College of Medicine, Tzu Chi University, Hualien 97004, Taiwan; yri100@gms.tcu.edu.tw (R.-I.Y.); wuws@mail.tcu.edu.tw (W.-S.W.); cordiallove@yahoo.com.tw (C.-C.C.); u8931246@mail.tcu.edu.tw (J.-R.W.); 2Institute of Medical Sciences, Tzu Chi University, Hualien 97004, Taiwan; smay12252001@yahoo.com.tw; 3School of Chinese medicine, China Medical University, Taichung 40402, Taiwan; u9922001@cmu.edu.tw; 4Division of Gastroenterology, Department of Medicine, Buddhist Tzu Chi General Hospital and Tzu Chi University, Hualien 97004, Taiwan; 5Research Centre for Hepatology, Buddhist Tzu Chi General Hospital, Hualien 97004, Taiwan

**Keywords:** Ling-Zhi-8, c-Met, EGFR, *N*-glycan, HCC

## Abstract

The poor prognosis of hepatocellular carcinoma (HCC) is resulted from tumor metastasis. Signaling pathways triggered by deregulated receptor tyrosine kinases (RTKs) were the promising therapeutic targets for prevention of HCC progression. However, RTK-based target therapy using conventional kinase-based inhibitors was often hampered by resistances due to compensatory RTKs signaling. Herein, we report that Ling-Zhi-8 (LZ-8), a medicinal peptide from *Ganoderma lucidium*, was effective in suppressing cell migration of HCC413, by decreasing the amount and activity of various RTKs. These led to the suppression of downstream signaling including phosphorylated JNK, ERK involved in HCC progression. The capability of LZ-8 in targeting multiple RTKs was ascribed to its simultaneous binding to these RTKs. LZ-8 may bind on the *N*-linked glycan motif of RTKs that is required for their maturation and function. Notably, pretreatment of the *N*-glycan trimming enzyme PNGase or inhibitors of the mannosidase (*N*-glycosylation processing enzyme), kifunensine (KIF) and swainsonine (SWN), prevented LZ-8 binding on the aforementioned RTKs and rescued the downstream signaling and cell migration suppressed by LZ-8. Moreover, pretreatment of KIF prevented LZ-8 triggered suppression of tumor growth of HCC413. Our study suggested that a specific type of *N*-glycan is the potential target for LZ-8 to bind on multiple RTKs for suppressing HCC progression.

## 1. Introduction

Hepatocellular carcinoma (HCC) is one of the most devastating cancers worldwide [1]. The poor prognosis and high recurrence rate of HCC are owing to tumor metastasis. Within the tumor microenvironment, numerous growth factors and cytokines can induce metastatic changes, including epithelial mesenchymal transition (EMT), and migration and invasion of the primary tumor [2]. Blocking the molecular pathway mediated by these factors is a promising strategy for suppressing tumor progression. A lot of metastatic factors including hepatocyte growth factor (HGF) [3] and epidermal growth factor (EGF) [3,4,5,6,7], can trigger the progression of HCC. Overexpression or uncontrolled activation of receptor tyrosine kinases (RTKs) of HGF and EGF, c-Met and EGFR [8,9], respectively, has been closely associated with HCC progression. Targeting these RTKs for prevention of HCC progression has been the subject of intensive study for decades. One unresolved issue for RTK target therapy is drug resistance [8,10,11,12,13] due to co-expression of multiple growth factors that may rise compensatory secondary signaling after treatment with specific tyrosine kinase inhibitors (TKIs) [14]. 

One of the potential therapeutic strategies for reducing drug resistance is to comprehensively blockade multiple RTKs by suppressing the *N*-glycan motif on RTKs. *N*-linked glycosylation is a critical posttranslational modification step in the maturation of transmembrane RTK glycoproteins [15,16]. *N*-glycans are also essential for maintaining the residency time and surface levels of multiple RTKs [16,17]. Most growth factor receptors, including EGFR [16,17], are *N*-glycosylated. Moreover, the role of *N*-glycans on RTK in tumor progression is emerging [16]. Thus, an anticancer agent capable of simultaneously targeting the universal *N*-glycans on oncogenic RTKs can be used to effectively suppress tumor progression [18]. Among the anticancer agents, the medicinal peptide LZ-8, which is a fungal immunomodulatory protein (FIP) that is purified from *Ganoderma lucidium* [19], has been highlighted. Previously, FIP were known to be capable of binding cell-surface proteoglycan to regulate immune function [20]. Recently, LZ-8 has been demonstrated to induce cell cycle arrest in lung cancer [21,22] and suppress cell migration of cervical cancer [23]. Our recent studies demonstrated that LZ-8 blocked c-Met dependent and independent signaling to achieve an anti-HCC progression [24] and suppressed gene expression of hydrogen peroxide clone-5 (Hic-5), one of the critical effectors for sustained reactive oxygen species (ROS) and mitogen activated protein kinase (MAPK) signaling [25]. 

In the present study, we found that LZ-8 was highly efficient at suppressing multiple RTKs, including EGFR, c-Met, and HER3, resulting in the blockage of downstream signaling and decreased cell migration. This may be attributable to the direct binding of LZ-8 to specific types of *N*-glycan on multiple RTKs.

## 2. Results

### 2.1. Suppression of HCC Migration by LZ-8

Recently, a patient-derived, highly motile HCC, HCC413, was established in the Research Centre for Hepatology, Buddhist Tzu Chi General Hospital, Hualien. Migration of HCC413 as well as that of another motile HCC, HCC329 [24,25], was suppressed dramatically by LZ-8 (Figure 1A,B). A transwell migration assay was also conducted for HCC413 (Figure 1C, left panel) and PLC5 [17], other HCC cell lines frequently used to study HCC progression (Figure 1C, right panel). LZ-8 also significantly suppressed cell growth of HCC413 by 25–30% using MTT method (Appendix A). 

### 2.2. LZ-8 Suppressed the Expression and Activation of Multiple RTKs in HCC

To investigate whether LZ-8 affected multiple RTKs that were responsible for HCC413 migration, we performed RTK receptor array analysis (using a kit from R&D Systems that could detect 49 active RTKs). Figure 2A demonstrates that several RTKs, especially EGFR and c-Met, were active in untreated HCC413. Notably, the activities of EGFR and c-Met were heavily suppressed by LZ-8 compared with the untreated group, whereas the other active RTKs (e.g., insulin receptor) were not significantly altered. Furthermore, in another RTK array (obtained from Cell Signaling Technology that could detect 18 active RTKs), two more active RTKs—HER3 (of the EGFR family) and c-Kit (of the PDGF family)—were also identified to be decreased by LZ-8 (Figure 2B). 

Western blot verified that phosphorylated HER3 (at Tyr 1298, p-HER3) was suppressed by LZ-8 (0.2 and 0.8 μM) by 60–85% in a dose-dependent manner at 24 h, whereas phosphorylated c-Met (at Tyr 1234, p-Met) was suppressed by 0.8 (but not 0.2) μM LZ-8 by 85% (Figure 3A,B). Consistently, total HER3 and EGFR were also diminished by 0.2 and 0.8 μM LZ-8 by 70–98%, whereas total c-Met was decreased by 0.8 (but not 0.2) μM LZ-8 by 90% (Figure 3A,B). In an immunoprecipitation of EGFR coupled with Western blot of phosphorylated EGFR (at Tyr1173, p-EGFR) analysis, LZ-8 suppressed total and phosphorylated EGFR by 52–55% (Figure 3C). In addition, phosphorylated c-Kit (at Tyr 703) was marginally suppressed by LZ-8 (data not shown). All together, these suggested that LZ-8 decreased both the amount and activities of multiple RTKs in HCC413. Moreover, LZ-8 suppressed the expression of EGFR and Her-3 and phosphorylation of HER3 in PLC-5 (Appendix A, compare lanes 1 and 2). For comparison, HepG2 which is a HCC cell line with low migratory ability [25], was also included. As shown in Appendix A, although HER3 was expressed and significantly suppressed by 0.8 μM LZ8 in HepG2, p-HER3 and EGFR could not be detected. In addition, FIP-vvo, a fungal immunoprotein (FIP) with 60% similarity with LZ-8 [20], suppressed phosphorylated HER3 and c-Kit by 80% and 45%, respectively (Appendix A). 

### 2.3. LZ-8 Suppressed Downstream Signaling in HCCs

Deregulation of EGFR [26,27] and c-Met [28] may trigger similar downstream signaling, such as Ras-MAPK and PI3-AKT for tumor progression. These signal cascades also mediated the molecular pathways initiated from HER3 [29,30]. Thus, we investigated whether LZ-8 suppressed the common downstream effectors, including JNK, AKT, and ERK [31,32], known to be involved in HCC progression. As shown in Figure 3D, phosphorylated JNK (p-JNK) was decreased by treatment of HCC413 with 0.2 and 0.8 μM LZ-8 for 24 h by 50% and 25%, respectively, whereas phosphorylated ERK (p-ERK) was decreased by 85–90% by LZ-8 at both concentrations. However, p-JNK and p-ERK were not observed in PLC-5, and phosphorylated AKT was not detected in both HCC413 and PLC-5 (data not shown). Thus, the signal pathway downstream of RTK in PLC-5 requires further investigation. In the nucleus, c-jun is the well-known transcriptional factor activated by JNK. Consistently, 0.2 and 0.8 μM LZ-8 suppressed phosphorylated c-jun (p-c-jun) in HCC413 by 75% and 50%, respectively, for 24 h in a pattern similar to that of p-JNK (Figure 3D,E). The total JNK, ERK, and c-jun (Figure 3D,E) were not decreased by LZ-8 (Figure 3A,B). By contrast, p-c-jun was low and not influenced by LZ-8 in PLC5, which is consistent with the absence of p-ERK and p-JNK (data not shown). We also examined whether LZ-8 affected upstream signaling components such as ROS and the focal adhesion adaptor Hic-5 known to positively cross talk with JNK-ROS signaling in the HCC [25]. As expected, LZ-8 suppressed ROS generation at 24 h and 48 h by 60–70% (Figure 3F, upper panel), and reduced Hic-5 expression by 70–80% for 24 h (Figure 3D,F) in HCC413. Moreover, LZ-8 significantly suppressed phosphorylation of FAK (Figure 6C), a focal adhesion kinase that is critical for mediating HCC progression [33]. Furthermore, FIP-vvo also suppressed ROS generation and Hic-5 expression by 75% and 60%, respectively, in HCC413 (Figure 3F, left and right panel). By contrast, basal levels of Hic-5, p-ERK, p-JNK were not observed in HepG2 (Appendix A). Although basal p-c-jun was high in HepG2, it could not be suppressed by LZ-8 and was even induced by LZ-8 in a dose-dependent manner (Appendix A).

We further investigated whether LZ-8 could block ligand-induced RTK signaling in HCC340, another patient-derived HCC responsive to HGF [15]. As depicted in Appendix A, LZ-8 prevented HGF-induced phosphorylation of c-Met, ERK, and c-jun, by 95%, 30%, and 90%, respectively, at 24 h. However, prevention of HGF-induced JNK phosphorylation by LZ-8 was not observed. LZ-8 also prevented EGF- and TGF-β-induced c-jun phosphorylation at 24 h (Appendix A) by 95% and 40%, respectively. Consistently, LZ-8 greatly suppressed the EGF- and HGF-induced cell migration of HCC340 (Figure 1B). Moreover, LZ-8 not only decreased the constitutive migration of HCC340 but also completely prevented the enhanced migration of HCC340 that resulted from the overexpression of EGFR (Figure 1D). Elevation of EGFR in HCC340 transfected with EGFR-expressing plasmids was validated by Western blot (Appendix A). In addition, whereas overexpression of EGFR did not increase HCC340 cell growth, LZ-8 could decrease growth of parental HCC340 within 48 h (data not shown).

### 2.4. LZ-8 Display Colocalization with EGFR and c-Met in Confocal Immunofluorescent Analysis

To investigate whether the aforementioned RTKs are direct targets of LZ-8 in HCC, we examined whether they could be colocalized with LZ-8 in HCC413 treated with FITC-labeled LZ-8 using confocal immunofluorescent analysis. As displayed in Figure 4, colocalization of cell surface EGFR (Figure 4A, upper panel), HER3 (Figure 4B, upper panel), and c-Met (Figure 4C, upper panel) with LZ-8 was evident after treatment with FITC-labeled LZ-8 for 5 min. In addition, colocalization of the RTKs with FITC-labeled LZ-8 was also apparent in PLC-5 (Appendix A). Because both LZ-8 and RTK signals were relatively high in both HCCs, two more experiments were performed to address this issue. First, when HCC413 (Appendix A, left panel) and PLC-5 (Appendix A) were treated with cycloheximide (CHX) for 6 h, membrane and cytosol RTK were dramatically decreased while most of the staining in perinuclear regions were retained. Moreover, in a control image obtained by staining HCC413 with secondary antibody only, no membrane and cytosolic staining was observed while minimal background remained in the perinuclear region (Appendix A, right panel). Taken together, both membrane and cytosol staining in the confocal immunofluorescence of EGFR and HER3 were actually derived from each RTK protein, whereas the staining around the perinuclear region represented background interference. On the other hand, the intracellular processing of LZ-8 was analyzed in a time-dependent manner in HCC413 using LZ-8 primary antibody followed by secondary antibody labeled with FITC. As indicated in Appendix A, in the cells treated with LZ-8 for 5 min, most of the FITC fluorescence was observed near the cell surface. After 10–45 min, the intracellular LZ-8 gradually increased, and much of LZ-8 localized at the perinuclear zone. This implies that LZ-8 may enter an endocytic pathway upon binding with RTK. The interactions of LZ-8 with EGFR on the cell membrane were investigated by IP EGFR/Western blot LZ-8 analysis. As shown in Appendix A, significant binding of EGFR with LZ-8 in membrane fractions of the LZ-8-treated cells could be observed at 5 min. However, the binding of LZ-8 with membrane EGFR decreased at 24 h. This finding further supports the notion that endocytosis of LZ-8 occurs after engagement with membrane RTK. 

We further examined whether LZ-8 interfered with the ligand binding of EGFR and c-Met by using flow cytometry. As demonstrated in Figure 5A, treatment of FITC-labeled EGF (EGF-FITC) and FITC-labeled HGF (HGF-FITC) revealed a time-dependent increase of binding on HCC413 from 5 to 30 min. Notably, the bindings of EGF-FITC and HGF-FITC on the cell surface were suppressed by unlabeled LZ-8 at 15 min by 60% and 30%, respectively (Figure 5B).

### 2.5. Mannosidase Inhibitors Prevented LZ-8 Binding on EGFR/c-Met

To investigate whether *N*-glycan motifs are involved in the binding of LZ-8 on RTK, PNGase, an enzyme capable of trimming all types of *N*-glycan, was employed. In a flow cytometry analysis, pretreatment with PNGase (100 U/mL), but not α-glucosidase or mannosidase for 4 h significantly decreased the surface binding of LZ-8 (from 22.39% to 9.6%) but not the surface binding of EGF on HCC413 (Appendix A), indicating that the *N*-glycan on the cell surface is essential for the binding of LZ-8. We further examined whether PNGase rescued the LZ-8-inhibited RTK expression and signal transduction in both HCC413 and PLC-5. As shown in Appendix A, pretreatment with PNGase (100 U/mL) but not α-glucosidase for 4 h significantly prevented the suppressive effect of LZ-8 on the expression of EGFR, HER3, and c-Met as well as the phosphorylation of ERK. However, PNGase could not rescue LZ-8-triggered suppression of EGFR and HER3 in PLC-5 (data not shown).

*N*-glycosylation of surface glycoprotein is processed in the endoplasmic reticulum and Golgi apparatus. Specifically, mannosidase is required to trim high-mannose precursor for the subsequent terminal addition of fucose and sialic acid, forming the complex type of *N*-glycan. The mature EGF receptor has been found to contain both complex- and high-mannose-type *N*-linked oligosaccharides in the approximate ratio of 2:1 [34]. Accordingly, we further investigated whether the type of *N*-glycan on RTK was critical for LZ-8 targeting. Specifically, we examined whether treatments with inhibitors of mannosidase, which favor the formation of high-mannose instead of complex *N*-glycans on RTK [15,16], could perturb the targeting of LZ-8 to RTKs. For this purpose, swainsonine (SWN), the inhibitor of Golgi apparatus alpha mannosidase II [15,16]; kifunensine (KIF), the inhibitor of endoplasmic reticulum mannosidase I [34] and deoxymannojirimycin (DMJ; Man8GlcNAc2 isomers), the inhibitor of glucosidase I, II, and mannosidase I [15], were used. As depicted in Figure 5C, flow cytometry analysis revealed that pretreatment with DMJ, KIF, and SWN for 24 h suppressed the bindings of LZ-8 on the cell surface of HCC413 by 60%, 70%, and 85%, respectively. Subsequently, we examined whether the colocalizations of LZ-8 with EGFR and HER3 on the cell surface of HCC413 could be interfered by KIF and SWN using confocal immunofluorescent imaging. Remarkably, in the cells pretreated with KIF and SWN for 24 h, the colocalization of FITC-labeled LZ-8 with cell surface EGFR and HER3 were 90–95% diminished (Figure 4A,B, middle and low panels, Figure 4D), which contrasted with the colocalization of LZ-8 and RTKs observed in the cells without pretreatment (Figure 4A,B upper panel, Figure 4D). However, the colocalization of LZ-8 with c-Met was prevented by pretreatment of SWN only (Figure 4C lower panel, Figure 4D). In the KIF-pretreated group (Figure 4C, middle panel), some colocalizations of LZ-8 and c-Met were still noted. To ascertain whether KIF was able to prevent the binding of c-Met with LZ-8, IP c-Met followed by Western blot of LZ-8 was performed. As shown in Figure 6A (upper panel), pretreatment with either SWN or KIF for 24 h significantly inhibited the binding of LZ-8 with EGFR by 40–50% in HCC413. The same preventive effect of both inhibitors on the binding of LZ-8 with c-Met could also be observed in the IP c-Met followed by Western LZ-8 analysis (Figure 6A, lower panel). The molecular weights of EGFR and c-Met in the KIF-pretreated cells were slightly lower than those in the non-pretreated cells, probably because the decreased mannosidase activity changed the *N*-glycan from the complex to high-mannose type. In the SWN-pretreated cells, the amount of EGFR and c-Met clearly decreased, thus resulting in the reduced binding of LZ-8. 

To substantiate the proposition that the alteration of *N*-glycan on RTKs (induced by the mannosidase inhibitor such as KIF) may affect the direct binding of LZ-8 with RTKs, an affinity blot using biotin-labeled LZ-8 as a probe was performed. As evident in Figure 6B, direct binding of biotin-labeled LZ-8 with immunoprecipitated EGFR, HER3, and c-Met from cells treated with KIF markedly decreased (by 40–80 %) in comparison with those from untreated cells. 

### 2.6. Mannosidase Inhibitors Rescued LZ-8 Suppressed Signaling, Cell Migration, and Tumor Growth

We further examined whether pretreatment of mannosidase inhibitors also influences RTK-dependent signaling suppressed by LZ-8. As demonstrated in Figure 6C, KIF significantly rescued the LZ-8-triggered suppression of RTKs, including HER3 and EGFR by 60–70% in HCC413 treated with LZ-8 for 24 h, whereas SWN rescued the LZ-8-triggered suppression of HER3 and EGFR by 75% and 15%, respectively. Furthermore, p-HER3 was rescued by KIF (but not SWN) by 85%. Regarding downstream signaling, KIF rescued LZ-8-triggered inhibition of phosphorylation of ERK, JNK, c-jun, and FAK as well as expression of Hic-5 by 85–95%. By contrast, SWN only slightly prevented the decreases of phosphorylation of ERK and p-c-jun by 10% but did not affect the level of other downstream signaling molecules (Figure 6C,D). In general, a single treatment with both inhibitors did not greatly affect the level of the aforementioned signaling molecules with the exception that KIF slightly suppressed phosphorylation of HER3 and SWN significantly decreased phosphorylation of ERK, JNK, and c-jun (Figure 6C). Currently, SWN treatments are being optimized in search of a suitable SWN concentration that is low enough not to suppress basal-level signaling and high enough to reverse LZ-8 inhibitory effects. However, we noted that LZ-8-triggered suppression of EGFR, HER3, and p-HER3 was not prevented by KIF and SWN in PLC-5 cells (Appendix A).

Subsequently, we examined whether mannosidase inhibitors affected the cellular effect of LZ-8 on HCC. As shown in Figure 1C, pretreatment of KIF, but not SWN, prevented the suppressive effect of LZ-8 on cell migration of HCC413 and PLC-5 by 80% and 95%, respectively, compared with the LZ-8-only group. In addition, a single treatment of KIF and SWN suppressed cell migration of HCC413 (Figure 1C, left panel) and PLC-5 (Figure 1C, right panel) by 15–20% compared with the control. The reversal of LZ-8-suppressed cell migration by pretreatment of KIF was also observed in another HCC, HCC340 (24) and A549, a frequently used lung cancer cell line, while single treatment of KIF had no significant effect on cell migration of HCC340 and A549 (Appendix A). Similarly, in a MTT assay, pretreatment of KIF also attenuated the LZ-8-triggered cell growth inhibition of HCC413, HCC340, and A549 (by 22%, 8%, and 13%), while single treatment of KIF had no significant effect (Appendix A). However, KIF did not prevent the LZ-8 suppressed cell migration (data not shown) and cell growth (Appendix A) of another HCC cell line, Hep3B. Moreover, we employed xenograft transplantation of HCC413 tumors in the livers of SCID mice as a model to determine whether KIF has preventive effects on LZ-8 in vivo. Notably, pretreatment with KIF also reversed the suppression of tumor growth triggered by LZ-8 by 70% (Figure 7A,B). On the molecular level, dramatic decreases of the key RTKs, EGFR, and c-Met, and one of the downstream signal molecule p-JNK were observed in tumors from mice treated with LZ-8 compared with those in the tumor from untreated (con) group (Figure 7C). Remarkably, in the tumor from mice treated with LZ-8 coupled with KIF, p-JNK and c-Met were much higher and EGFR was significantly higher than those of the LZ-8 alone group (Figure 7C). Western blot of LZ-8 also verified the distribution of LZ-8 in tumor cell from mice treated with LZ-8 and LZ-8/KIF but not the control group. Together, these results are consistent in demonstrating the reversal effect of KIF on the suppressive effects of LZ-8 in HCC413 in vitro.

## 3. Discussion

Deregulation of RTK correlates with the development of numerous cancers and highlights an attractive therapeutic strategy targeting oncogenic RTKs [35]. However, multiple oncogenic RTKs are often co-activated in tumor cells, which complicates the RTK-based target therapy [36,37,38,39,40]. Previously, we found LZ-8-suppressed tumor progression of HCC in a c-Met-dependent and -independent manner [24,25]. In the present study, several oncogenic RTKs, including EGFR, c-Met, HER3, and c-Kit (Figure 2 and Figure 3), were found to be co-activated in HCC in the patient-derived hepatoma cells, HCC413. Moreover, LZ-8 could simultaneously target and suppress the amount and activity of EGFR, c-Met, and HER3 (Figure 3A–C), leading to the downregulation of common downstream signaling molecules such as p-ERK, p-JNK, and p-c-jun in HCC413 (Figure 3D,E). This highlights how LZ-8 can be an effective anti-HCC agent able to target multiple RTKs that might overcome resistance due to compensatory signaling.

The underlying mechanism through which LZ-8 targets multiple RTKs is intriguing. We contend that LZ-8 is prone to bind on cell surface *N-*glycan motif of RTK, as evidenced by reversals of LZ-8-triggered molecular and cellular effects by antagonists of α-mannosidases involved in *N*-glycan processing (Figure 4, Figure 5 and Figure 6). Specifically, α-mannosidases play crucial roles in trimming and extension steps in the biosynthesis of *N*-glycans that determine the final nature of the mannose-containing oligosaccharides on transmembrane glycoprotein [15]. Suppression of α-mannosidase activity by specific inhibitors such as SWN and KIF may result in alteration of the type of *N*-glycans [15]. For example, after cells were incubated with SWN there was a dramatic decrease in the complex type of glycopeptides accompanied with an increase in the high-mannose type. Furthermore, KIF, in keeping with its inhibition of mannosidase I, caused a complete shift in the structure of the *N*-linked oligosaccharides from complex chains to the high-mannose Man9 (GlcNAc)2 structures. In our results, we found SWN and KIF effectively suppressed the colocalization and binding of LZ-8 with EGFR, c-Met, and HER3 by confocal immunofluorescence and IP/Western analysis (Figure 4 and Figure 6, respectively), although the preventive effect of KIF on colocalization of LZ-8 with c-Met requires further confirmation by confocal immunofluorescence. Thus, we propose that LZ-8 binding on both RTKs via complex type *N*-glycans (but not high-mannose type) is a topic worthy of further investigation through carbohydrate sequencing.

Recently, the role of *N*-glycans in tumor progression has emerged [16,41,42,43] and interference with the *N*-linked glycosylation pathway has been proposed as a novel strategy for suppressing cancer progression. For example, inhibiting *N*-linked glycosylation markedly reduces RTK signaling, sensitizing tumor cells to cytotoxic therapies. This approach has the advantage of blocking not only the primary, dominant signaling RTK (e.g., EGFR), but also the secondary signaling of co-expressed RTKs, thus alleviating resistance due to compensatory signaling [44]. However, blockade of *N*-linked glycosylation pathway raises concerns regarding toxicity induced by inhibitors of *N*-linked glycosylation. Our study reveals another method for interfering with the *N*-linked glycan on RTKs. Specifically, we suggest that bindings of LZ-8 with the *N*-glycan motif on EGFR, HER3, and c-Met are responsible for suppression of downstream signaling and cell migration (Figure 5 and Figure 6). This is consistent with a recent study that demonstrated that surfactant protein D, a member of the endogenous collectin family in the lung, inhibited the proliferation, migration, and invasion of A549 human lung adenocarcinoma cells by reducing EGFR signaling via interacting with the *N*-glycans motif [45].

In addition to LZ-8, we also found that FIP-vvo, the FIP that is a 60% homolog with LZ-8, was capable of targeting multiple RTKs. Most FIPs target the carbohydrate motif of surface glycoprotein such as *N*-glycan on RTK via a specific binding domain such as putative carbohydrate-binding modules (CBMs), as has been found in the FIP-vvo protein C-terminal region [25]. Specifically, a CBM-34 structure is responsible for the interaction between glycoprotein on the cell surfaces of hPBMCs and FIPs. Studies have revealed the structure domain of LZ-8, raising a possibility that CBMs in LZ-8 can be defined. Accordingly, it may be possible to develop CBMs-containing LZ-8 mutants with a more efficient *N*-link glycan targeting capability than parental LZ-8 using structure modification coupled with drug screening approaches.

## 4. Materials and Methods

### 4.1. Cell Lines and Chemicals

HCC413, PLC-5, HCC340, HepG2, Hep3B, and A549 cells were cultured in Dulbecco’s Modified Eagle’s Medium with 10% fetal bovine serum. HCC413 and HCC340 are clinically derived HCC cell line established from a series of HCC tissues obtained from surgery with patients’ consent, the use of which was approved by the Research Ethic Committee of Buddhist Tzu Chi General Hospital (IRB 101–62), as described in a previous report [24]. HepG2 and Hep3B were gifts from Dr. Huang in TZU CHI University. PLC-5 was a gift from Yeastern Company (Taipei, Taiwan). Antibodies for each signaling molecule were obtained from Santa Cruz Biotechnology, Inc. (Santa Cruz, CA, USA). LZ-8 and rabbit LZ-8 primary antibody were purchased from the Yeastern Company (Taipei, Taiwan). Plasmids of EGFR (p-EGFR) were a kind gift from Dr. Hsu, Tsui-Ling of Academia Sinica (Taipei, Taiwan).

### 4.2. Wound Healing and Transwell Migration Assays, Immunoprecipitation, and Western Blot

The experiments were performed according to the methods described in our previous study [25].

### 4.3. Receptor Array Analysis

A receptor array kit (Proteome Profile™ Array; R&D System, Minneapolis, MN, USA) capable of simultaneously detecting 49 essential RTKs in phosphorylated form was employed for screening the phosphorylated RTK signaling in HCC413, as in our previous report [24]. The other receptor array kit (PathScan antibody Array, Cell Signaling Technology, Danvers, MA, USA) capable of detecting 18 RTKs in phosphorylated form was also used according to the manufacturer’s protocol.

### 4.4. FITC-Labeled Proteins

Each protein was reacted with fluorescein isothiocyanate following the user’s guide (Pierce FITC Labeling Kit; Thermo Scientific, Waltham, MA, USA). The excess fluorescent dye was removed by Slide-A-Lyzed MINI Dialysis Units (3, 500 MWCO; Thermo Scientific). The degree of labeling and the concentration of the FITC-labeled proteins were quantified by spectrophotometry at 280 nm and 495 nm. 

### 4.5. Confocal Microscopy for Colocalization of LZ-8 and Candidate RTK

Cells were fixed with 4% paraformaldehyde in PBS, followed by permeabilization with 0.1% Triton X-100, blocked with 5% bovine serum albumin (BSA) and then incubated with Abs against the indicated signaling molecules for 1 h at room temperature. The nuclei were counterstained with DAPI. The images of each molecules were observed with an FiX10 (Olympus) at 600× magnification. To verify colocalization, the two separate fluorescent objects were analyzed using Zen Imaging Software (ZEISS, Oberkochen, Germany). The threshold was adjusted to eliminate background noise. For quantification of cells exhibiting colocalization of the indicated molecules, the numbers of indicated colocalizations in four fields were scored and averaged. Each experiment was repeated and validated by two other investigators who did not know the treatment groups before microscopic observation.

### 4.6. Subcellular Fractionation

Briefly, the cells were suspended in hypotonic buffer followed by centrifugation at 2500 rpm for 3 min. The supernatants were then subjected to ultracentrifugation at 25,000 rpm for 1 h, and pellets were subsequently dissolved in lysis buffer containing 1% Triton X-100. Following a second centrifugation at 25,000 rpm for 1h, the supernatants were obtained as the membrane fraction.

### 4.7. Flow Cytometry

Cells were seeded on 24-well plates for 24 h. After appropriate treatments, the cells were fixed with ice-cold 1% paraformaldehyde for 30 min at 4 °C. The fixed cells were washed with 2 mL of washing buffer (0.5% BSA, 2 mM EDTA in PBS, pH 7.2) and flow cytometric analysis was performed using a Gallios flow cytometer (Beckman Coulter, Brea, CA, USA). Kaluza software version 1.2 (Beckman Coulter) was employed for mean fluorescence intensity data analysis. 

### 4.8. Establishment of Xenograft Liver Transplantation to Assess the Effect of LZ-8 in HCC Progression

The progression of HCC was established using NOD SCID mice as previously [25]. All the mice were males, 8 weeks old, and had an average body weight of 35–40 g. HCC413 cells (2 × 10^6^) were grown on the right flank of mice to obtain tumor mass (0.5 cm in diameter). The tumors (2 mm in diameter for each mouse) were then xenografted onto the right lobe of the SCID mice’s livers. After 3 days, the mice were intraperitoneally injected with the indicated agents (including LZ-8 alone or LZ-8 coupled with KIF group) twice a week. In a preliminary test, Western blot of LZ-8 proved that LZ-8 can be detected in each lobe of liver tissues after intraperitoneal injection for 3 h (Appendix A), validating the distribution of LZ-8 to liver tissues. Twenty to thirty days after undergoing transplantation and receiving the appropriate treatments, the mice were sacrificed to examine the primary tumor growth on the right lobes. Nodules with diameters exceeding 0.1–0.2 cm on the left or right lobes were denoted as secondary tumor foci. The animal experiment was approved by the Institutional Animal Care and Use Committee at Tzu Chi University, and regulations relevant to the care and use of laboratory animals were followed.

### 4.9. Statistical Analysis

A paired Student’s *t* test was conducted to analyze the differences in band intensities between samples on the Western blot and the differences in flow cytometry among the indicated samples. Quantitative data were expressed as mean ± coefficient variation, indicated by the error bars in each figure. Analyses of variance were performed to compare the differences in tumor masses between the different treatments.

## 5. Conclusions

LZ-8 blocked multiple oncogenic RTKs including EGFR, c-Met, and HER3 as well as downstream MAPK signaling for suppression of HCC413 migration. These can be reversed by the *N*-glycan trimming PNGase and inhibitor s of α-mannosidase KIF and SWN, suggesting that *N*-glycan, possibly the complex/hybrid type, is the critical structural motif on the oncogenic RTKs targeted by LZ-8. This proposed model is schemed in Figure 8.

## Figures and Tables

**Figure 1 cancers-11-00009-f001:**
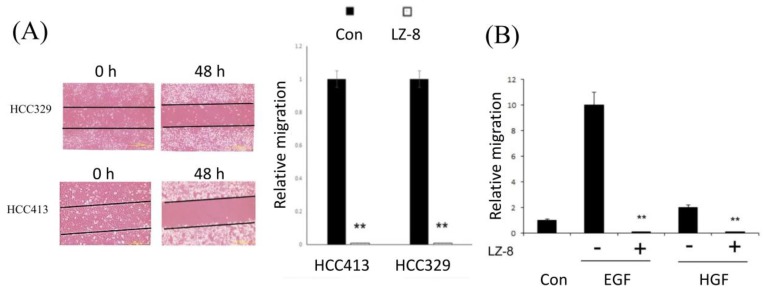
LZ-8 suppressed constitutive and growth factor-induced hepatocellular carcinoma (HCC) cell migration which can be rescued by mannosidase inhibitor. (**A**) HCC329 and HCC413 were untreated (control) or treated with LZ-8 (0.2 μM) for 48 h. (**B**) HCC340 cells were untreated (con), treated with 25 nM HGF or EGF (0.75 nM) alone, or treated with each growth factor coupled with LZ-8. (**C**) HCC413 and PLC-5 cells were untreated (Con), pretreated with KIF or SWN (5.8 μM) for 24 h followed by treatment with or without LZ-8 for 48 h. Transwell migration assay were performed. In (**B**) and right panel of (**A**), relative migration of HCC was calculated, taking the data for each control cell as 1.0. (**) represents the statistically significant difference (*p* < 0.05) between the LZ-8 treated vs. untreated group (**A**) and LZ-8 coupled with indicated growth factor vs. growth factor only group (**B**), n = 6. The images in (**C**) are representative of two reproducible experiments. (**D**) HCC340 cells seeded on a transwell migration culture cup were transfected with p-CMV vector or EGFR-expressing plasmid for 48 h. The cells transfected with EGFR plasmid were then treated with LZ-8 or left untreated for 48 h before migration assays were performed. Data are the average of two reproducible results.

**Figure 2 cancers-11-00009-f002:**
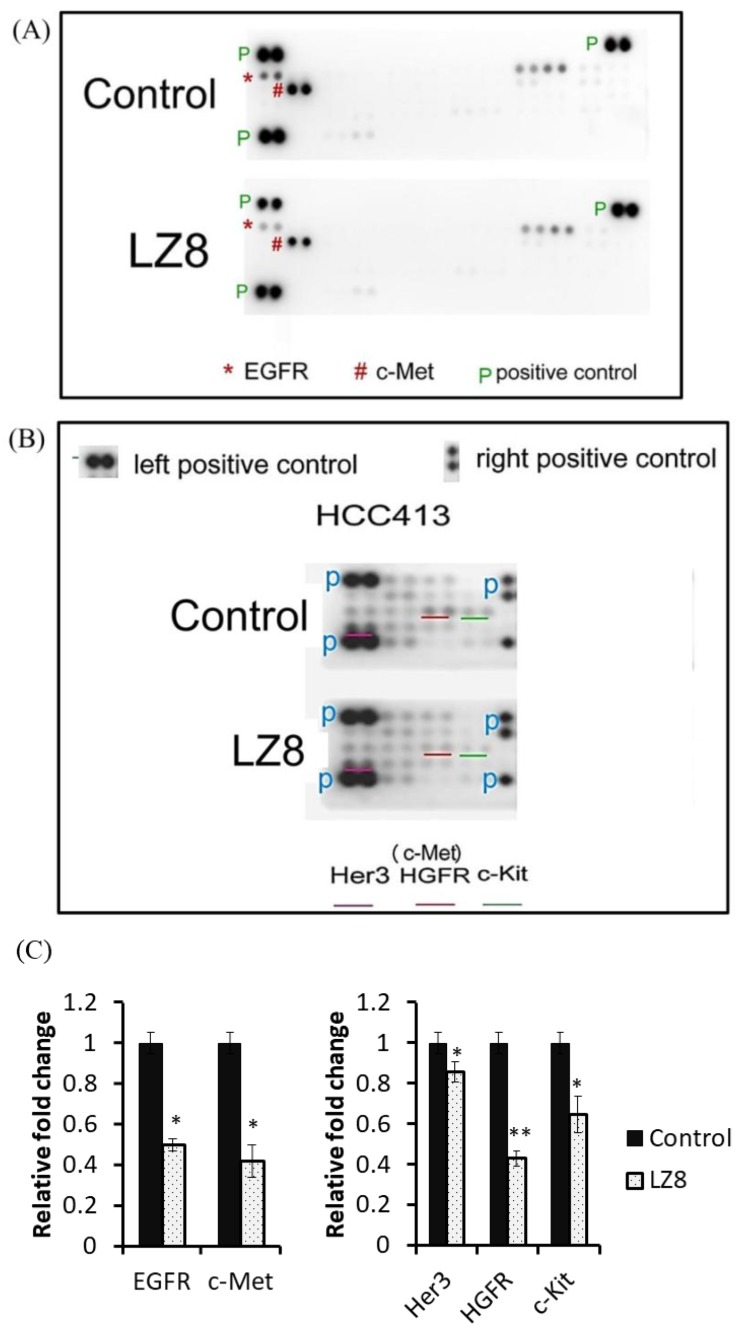
Receptor array analysis to detect RTK signaling in HCC413 with or without LZ-8 treatment. (**A**,**B**) Total cell lysate (300 μg/mL) of untreated (control, upper panel) and LZ8-treated (lower panel) HCC413 was used to analyze the relative amounts of 49 and 18 phosphorylated RTKs by receptor array analysis using kits from R&D Systems and Cell signal Technology, respectively, as described in the Materials and Methods section. The symbol *, # and underlines colors in (**A**,**B**), respectively, highlighted the RTKs inhibited by LZ-8 compared with those in the control group. The blue and green letters (P) indicate the positions of the positive controls included on the receptor array. (**C**) Quantification of relative fold change of indicated molecules were calculated as described in Materials and Methods. The ratios of the LZ8 vs. control, taking the data of the untreated group (control) as 1.0, (**, *) represents the statistically significant difference (*p* < 0.01, *p* < 0.05) of the indicated molecules between the LZ-8 treated and untreated group, n = 2.

**Figure 3 cancers-11-00009-f003:**
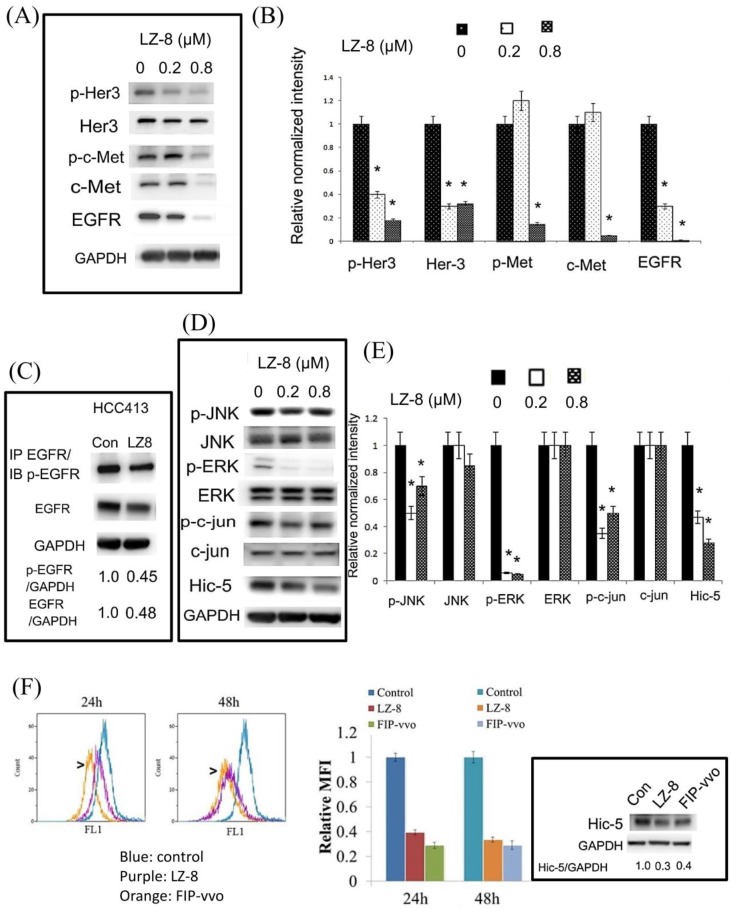
LZ-8 suppressed expression of RTK and downstream signaling. (**A**,**D**) HCC413 were treated with the indicated concentration of LZ-8 for 24 h. Western blots of the indicated molecules were performed using GAPDH as an internal control. (**B**,**E**) are the quantitative figures of (**A**,**D**), respectively. Relative normalized intensities of indicated molecules were calculated as the ratios of the indicated molecule vs. GAPDH, taking the data of the untreated group (zero concentration) as 1.0. (*) represents the statistically significant difference (*p* < 0.05) of the indicated molecules between the LZ-8 treated and untreated group, n = 4. (**C**) HCC413 were untreated (con) or treated with LZ-8 (0.2 μM). IP of EGFR followed by Western blot of phosphorylated EGFR (p-EGFR) and unphosphorylated EGFR were performed. GAPDH was employed as a loading control. The numbers are the ratio of the indicated molecules vs. GAPDH. Data are the average of three reproducible experiments. (**F**) HCC413 was untreated (control) or treated with LZ-8 or FIP-vvo for 24 h. ROS assay (left panel) and Western blot of Hic-5 (right panel) were performed. For ROS quantitation, the relative mean fluorescence intensity (MFI) of DCF-DA in the indicated treatments was calculated, taking the data of control as 1.0. The raw data of flow cytometry with a fluorescence distribution curve for HCC413 are presented in the left panel. In the Hic-5 Western blot, the numbers are the relative ratio of Hic-5 vs. GAPDH (as an internal control), taking the data of control as 1.0. Data are the average of two reproducible experiments.

**Figure 4 cancers-11-00009-f004:**
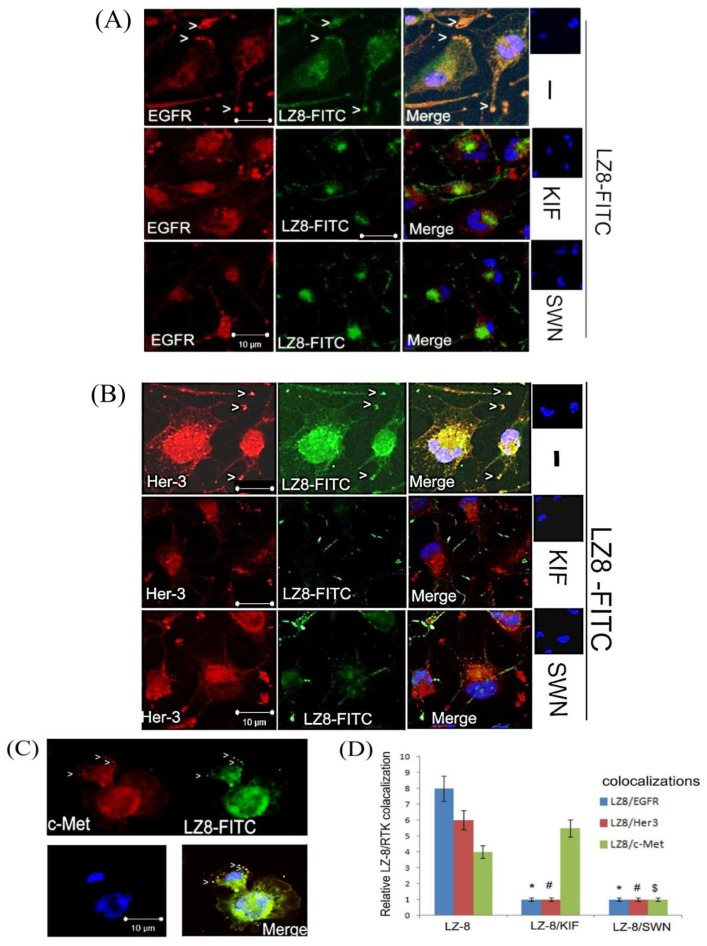
Direct bindings of LZ-8 with RTKs were suppressed by mannosidase inhibitors. (**A**–**C**) HCC413 was untreated (upper panel of **A**–**C**) or pre-treated with KIF (middle panel of **A**–**C**) or SWN (lower panel of **A**–**C**) for 24 h followed by treatment with FITC-labeled LZ-8 for 5 min. After fixation, immunostaining for (**A**) EGFR, (**B**) HER3, and (**C**) Met were performed, followed by confocal fluorescence analysis. The white arrow heads indicate colocalization of LZ-8 with the indicated RTKs depicted as yellow dots in the merge image. For each treatment, nuclear counterstaining is presented in the upper right small panel. (**D**) is the quantitative figure for LZ-8/RTK colocalization. The symbols (*, #, $) represent the statistically significant difference (*p* < 0.05) between the indicated sample with the LZ-8-only group. Data are the average of three reproducible experiments.

**Figure 5 cancers-11-00009-f005:**
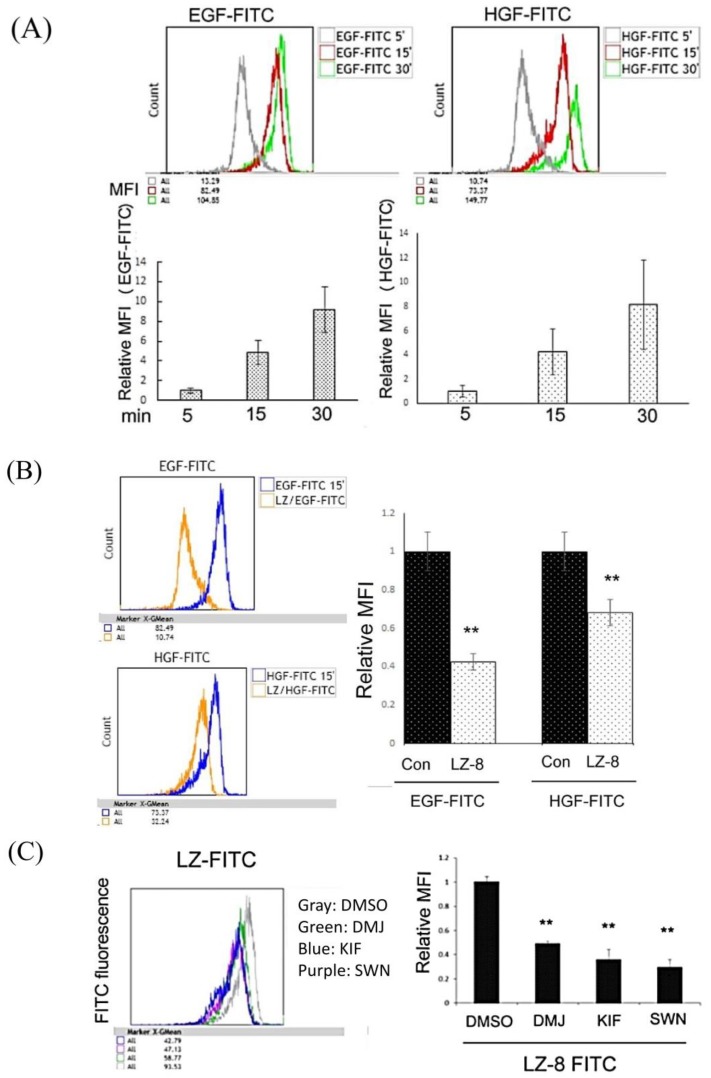
LZ-8 prevented the binding of EGF and HGF on the cell surface. (**A**) HCC413 cells were treated with EGF-FITC or HGF-FITC for the indicated times; (**B**) HCC413 cells were treated with EGF-FITC or HGF-FITC alone or each growth factor coupled with unlabeled LZ-8 for 15 min. (**C**) HCC413 cells were pre-incubated with the indicated mannosidase inhibitor for 16 h, followed by treatment with FITC-labeled LZ-8 (LZ-FITC) (100 ng/mL) for 15 min. Flow cytometry was performed. In the lower (**A**) and right (**B**) panel of each figure, the relative mean fluorescence intensity (MFI) is presented, taking (**A**) the 5 min group, (**B**) the EGF-FITC- or HGF-FITC-only group, or (**C**) LZ-FITC-plus-DMSO- or LZ-FITC-only group as 1.0. (**) represents the statistically significant difference (*p* < 0.05) between the control (Con) and LZ-8-treated groups (**B**) and DMSO and mannosidase inhibitors-treated groups (**C**), n = 3. In (**C**) KIF, Kifunensine (20 μM), SWN: Swainsonine (1 μg/mL).

**Figure 6 cancers-11-00009-f006:**
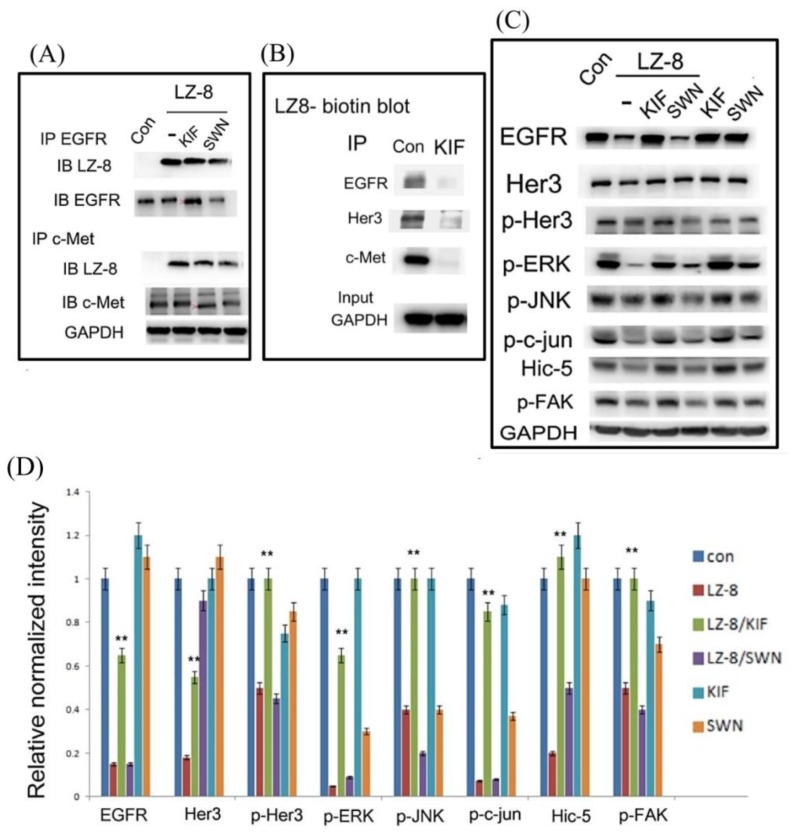
Binding of LZ-8 with RTKs and LZ-8-suppressed downstream signaling were prevented by mannosidase inhibitors. (**A**) HCC413 cells were untreated (con), or pre-incubated with the indicated mannosidase inhibitors for 16 h followed by treatment with LZ-8 for 30 min. IP of the indicated molecules followed by immunoblot (IB) of LZ-8 was performed using GAPDH as the input control. Arrowheads indicate the shift of EGFR (upper panel) and c-Met (lower panel) in the IP/IB analysis. (**B**) HCC413 cells were untreated (con) or treated with 10 μM KIF for 24 h, and IP of the indicated RTKs were performed followed by affinity blotting using biotin-labeled LZ-8 as a probe and avidin-HRP for chemiluminescent detection. Western blot of GAPDH was performed as an internal control. (**C**) HCC413 cells were untreated (con) or pre-incubated with the indicated mannosidase inhibitors for 16 h followed by treatment with LZ-8 or left untreated for 30 min. Western blot of the indicated molecules was performed using GAPDH as the internal control. (**D**) represents the quantitative figures for (**C**). Relative normalized intensities were calculated, taking the data of the untreated cells (Con) as 1.0. (**) represents the statistically significant difference (*p* < 0.05) between the indicated sample and the LZ-8-only group. KIF: Kifunensine (10 μM), or SWN: Swainsonine (5.8 μM).

**Figure 7 cancers-11-00009-f007:**
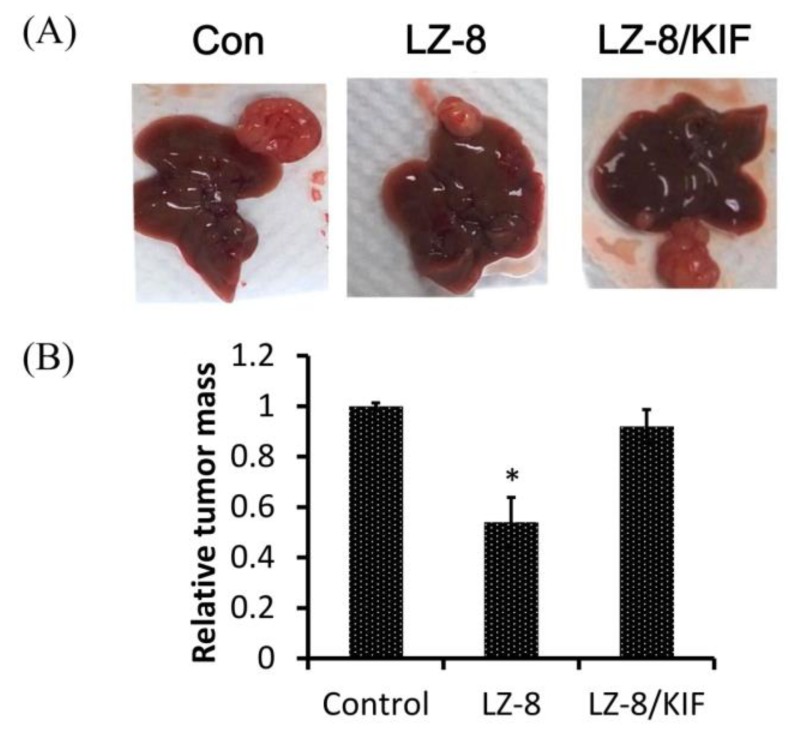
KIF prevented the suppression of HCC413 growth by LZ-8 in SCID mice livers. (**A**) HCC413 tumor was obtained from growths of inoculated HCC413 on the right flanks of SCID mice. The tumors were xenografted on the right lobe of SCID mice livers. After 3 days, the mice were intraperitoneally injected with LZ-8 (20 μg/mouse), phosphate buffered saline (PBS), or pretreated with KIF (10 μg/mouse) for 24 h followed by LZ-8 (20 μg/mouse) twice a week. The mice were sacrificed 20 days after tumor transplantation. Liver pictures were taken, and a representative image for each experimental group is shown. The location of the HCC413 tumor is indicated by arrows. (**B**) Quantitation of relative tumor weight from each mouse was calculated and averaged taking the data of control (PBS-treated) as 1.0. * represents the statistically significant difference of tumor weight between LZ-8 and the control group (*p* < 0.005, n = 3). (**C**) Tumor tissue from liver tumor of mice untreated (Con), treated with LZ-8 (40 μg/mice) alone and LZ-8 coupled with KIF (10 μg/mice) were subjected to extraction of lysate followed by Western blot of indicated molecules, GAPDH was used as an internal control.

**Figure 8 cancers-11-00009-f008:**
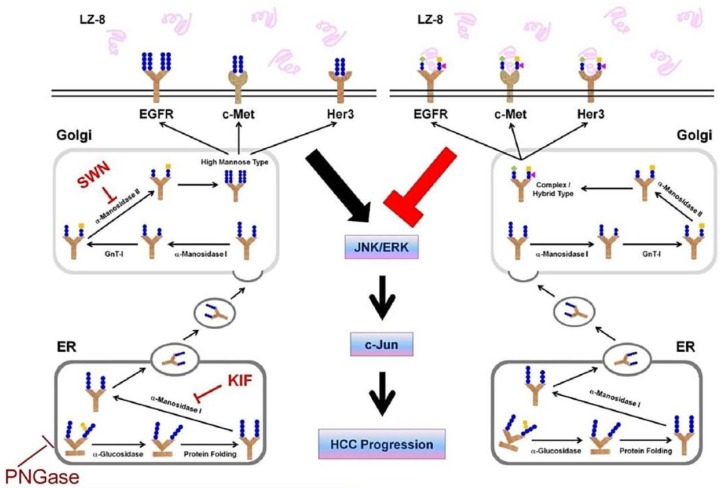
Proposed mechanisms by which LZ-8 suppresses HCC progression. After binding of LZ-8 with multiple oncogenic RTKs—including EGFR, c-Met, and HER3—via *N*-glycan, RTK activity, downstream MAPK signaling and HCC413 progression can be suppressed. These can be reversed by the *N*-glycan trimming PNGase and inhibitor of α-mannosidase, KIF, suggesting that *N*-glycan, possibly the complex/hybrid type, is the critical structural motif on the oncogenic RTKs targeted by LZ-8.

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
