# Peer review of "Involvement of N-glycan in Multiple Receptor Tyrosine Kinases Targeted by Ling-Zhi-8 for Suppressing HCC413 Tumor Progression"

_cancers, 2018, doi:10.3390/cancers11010009_

Round 1

Reviewer 1 Report

Peptides have multiple biological and pathological functions and displayed their importance in drug development with many peptide drugs on market currently. Ling-Zhi-8 (LZ-24 8) is a medicinal peptide purified from Ganoderma lucidium and displayed its ability in cell cycle arrest and cell migration in several cancers. Interestingly, authors in this manuscript demonstrated that LZ-8 suppressed HCC tumor growth and may act by binding on the N-linked glycan motif of

receptor tyrosine kinases (RTKs). It is very important and interesting study. However, there are questions I concern.

1, what is LZ-8 amino acid sequence? What molecular weight? What concentration (M) authors used? Or can author use Mol to replace mg/ml?

Concentrations such as 5 mg/ml and 10 mg/ml used may be too high because concentration used is very important to judge results.

2, Can or did authors use another peptide as positive or negative control?

3, Identical unit could be better. Authors used different units. Make readers confused (for example, LZ-8, 5mg/ml, KIF, 10uM, SWN, 10 ug/ml)

4, authors said that KIF could reverse the suppression of LZ-8 on tumor growth. Did authors test if KIF itself increased tumor growth?

5, Some reference may be wrong? For example, authors mentioned they did in reference no.15 that may possibly be other people’s work, not authors’ previous work. 

Reviewer 2 Report

In this study authors tried to prove that RTK signaling as the major target of LZ-8 in its intervention to HCC cells. Major concern is as follows,

1. The authors only validated the effect of LZ-8 with functional assays in multiple cell lines. It is not clear if the effect of LZ-8 is cell line specific.

2. In vivo study is not supportive effect. At the minimum detection on relevant signaling indicated in in vitro study should be performed.

3. It is quite interesting that LZ-8 can suppress tumour growth in vivo, but a lot of issue should be addressed. E.g., is LZ-8 distributed to the liver/tumour? Is LZ-8 target specific? This data may suggest the consistence between in vitro and in vivo observation about the mechanisms.

4. As the author can label LZ-8 with FITC, will they consider study the in vivo trafficking of the peptide?

5. Figure 4, the results are not fully supportive to your claim. Appropriate control assay shall be performed.

6 some minor points,

 a. the quality of image shall be improved.

 b. for the kinase array, quantification should be provided.

 c. the manuscript shall be checked by native speaker/English editing professionals.

Round 2

Reviewer 1 Report

this revision version is acceptable 

Reviewer 2 Report

Nil